# The Effectiveness of 360-Degree Virtual Reality-Based Mechanical Ventilation Nursing Education for ICU Nurses

**DOI:** 10.3390/healthcare13141639

**Published:** 2025-07-08

**Authors:** Doo Ree Kim, Jaeyong Yoo

**Affiliations:** 1College of Nursing, Konyang University, Daejeon 35365, Republic of Korea; kdr2015@konyang.ac.kr; 2Department of Nursing, College of Medicine, Chosun University, Gwangju 61452, Republic of Korea

**Keywords:** clinical reasoning, intensive care unit, learning immersion, mechanical ventilation, nursing education, self-efficacy, virtual reality

## Abstract

**Background/Objectives**: Mechanical ventilation management is a critical competency for intensive care unit (ICU) nurses; however, traditional training methods are often insufficient to prepare nurses for the complexities of alarm management and clinical decision-making. This study aimed to evaluate the effectiveness of a 360-degree virtual reality (VR)-based mechanical ventilation nursing education program for ICU nurses in Korea. **Methods**: A quasi-experimental pre-test–post-test design was employed with 65 ICU nurses (32 in the experimental group and 33 in the control group). Data were collected from May to October 2023. The VR-based program, developed using the ADDIE instructional design model, incorporated simulation-based scenarios focusing on ventilator alarm management and clinical reasoning. Outcome measures included knowledge of ventilation nursing, self-efficacy, clinical reasoning, learning immersion, turnover intention, and educational satisfaction. Data were analyzed using normality tests, descriptive statistics, independent *t*-tests, and paired *t*-tests. **Results**: The experimental group demonstrated significantly greater improvements in knowledge (Δ = 5.54), self-efficacy (Δ = 0.94), clinical reasoning (Δ = 0.76), and learning immersion (Δ = 0.88) compared to the control group (all *p* < 0.001), where Δ denotes the change score (post-test minus pre-test). Post-test assessments were conducted immediately after the intervention. Educational satisfaction was also significantly higher in the experimental group (*p* < 0.001). No significant difference was observed in turnover intention between the groups, suggesting a limited short-term impact on this outcome. **Conclusions**: A 360-degree VR-based education program effectively enhanced key competencies among ICU nurses. While these findings reflect short-term outcomes, future research is warranted to assess the long-term effects and sustainability of VR-based learning in ICU continuing education.

## 1. Introduction

The intensive care unit (ICU) is a dynamic and high-acuity clinical environment where nurses are required to provide highly specialized and complex care to critically ill patients [1]. Among the core competencies expected of ICU nurses, the ability to manage mechanically ventilated patients effectively is particularly essential [2,3]. Mechanical ventilation is a complex yet essential intervention in ICUs, necessitating that nurses interpret ventilator alarms and make timely, evidence-informed clinical decisions [4]. Inadequate management may lead to adverse outcomes, including ventilator-associated complications and increased patient mortality [5,6].

ICU nurses play a pivotal role in ensuring the safe and effective use of mechanical ventilation, which necessitates not only solid theoretical knowledge but also advanced clinical reasoning, critical decision-making, and psychomotor skills [1,7,8]. Traditional training methods—such as didactic lectures and mannequin-based practicums—often fail to replicate the complexity of ventilator alarm management and real-time clinical decision-making [3,6,9]. These educational limitations were further compounded during the COVID-19 pandemic, which restricted access to clinical learning environments and underscored the urgent need for rapid competency development among ICU nurses [9,10].

In Korea, a lack of high-fidelity simulation in ICU nurse education further highlights the need for innovative training approaches [11]. Virtual reality (VR) technology, particularly 360-degree immersive VR, offers a promising strategy to address this gap [12]. 360-degree VR refers to a computer-generated immersive environment where users can engage with a three-dimensional scenario from all directions, simulating a lifelike clinical experience. It typically involves stages such as scenario development, spatial mapping, and user interaction, which together support experiential learning in complex healthcare settings [13]. By simulating realistic ICU scenarios, VR enables nurses to engage in interactive, high-fidelity learning experiences that closely mirror clinical reality without compromising patient safety [12,14,15]. The existing literature suggests that VR-based education enhances various learning outcomes among nursing students and novice nurses, including self-efficacy, clinical reasoning, and learning immersion [10,16,17,18,19]. Learning immersion, defined as the degree to which learners experience engagement and flow during educational activities [20], has been identified as a key factor in promoting deep learning and skill acquisition in simulation-based education. For example, a previous study [10] demonstrated that a VR-based mechanical ventilation nursing education program significantly improved nursing students’ self-efficacy, clinical reasoning ability, and learning satisfaction. In a related study, Heo et al. [21] reported that novice nurses enhanced both their competencies and confidence through augmented reality-based mechanical ventilation setup training using a head-mounted display (HMD).

While previous studies have primarily focused on undergraduate nursing students or newly graduated nurses, our study expands the scope by targeting practicing ICU nurses. For instance, unlike Lee and Han [10], who examined nursing students using 2D VR content, our intervention employed a 360° three-dimensional, scenario-based VR training specifically designed to simulate real-life ventilator alarm situations in critical care settings. Additionally, we incorporated broader outcome variables—such as learning immersion and turnover intention—which have been rarely explored in VR-based nursing education research [13]. Turnover among ICU nurses is a global issue, with high rates of burnout, moral distress, and job dissatisfaction contributing to workforce instability [22]. This issue is particularly pertinent in Korean ICUs, where high patient acuity and staffing shortages further exacerbate turnover risk [23]. Strengthening ICU nurses’ competence and confidence through effective educational interventions may help mitigate the factors contributing to turnover intention and foster workforce retention and continuity of care. However, empirical evidence directly linking educational interventions—particularly VR-based programs—to reductions in turnover intention remains limited.

To address these gaps, this study developed and evaluated a 360° VR-based mechanical ventilation nursing education program tailored for ICU nurses in Korea, employing the ADDIE instructional design model for systematic content development and delivery. The program aimed to enhance ventilator management competence by improving ICU nurses’ skills in managing ventilator alarms, integrating patient assessments, and applying clinical reasoning in simulated ICU scenarios. Specifically, this study examined the program’s effects on the knowledge of ventilation nursing, self-efficacy, clinical reasoning ability, learning immersion, turnover intention, and educational satisfaction. We hypothesized that participation in the VR-based program would significantly improve knowledge, self-efficacy, clinical reasoning, and learning immersion, compared to conventional education, and may also positively influence turnover intention and educational satisfaction. By advancing the understanding of the effectiveness of 360° VR-based education in ICU nursing, this research seeks to inform future educational practices and contribute to professional development, competence retention, and workforce sustainability among ICU nurses in Korea.

## 2. Materials and Methods

### 2.1. Study Design

This study employed a quasi-experimental, non-equivalent control group pre-test–post-test design to evaluate the effectiveness of a 360° VR-based mechanical ventilation nursing education program for ICU nurses. The design was chosen to enable the comparison of outcomes between an experimental group exposed to the VR intervention and a control group receiving standard educational materials. Given the practical constraints of clinical settings and ethical considerations, random assignment was not feasible.

### 2.2. Study Participants and Setting

Participants were registered nurses currently working in the ICUs of a university-affiliated tertiary hospital in G city, South Korea, with less than three years of total clinical nursing experience. Four ICUs—medical, surgical, neurological, and emergency—were included, each with an average capacity of approximately 20 beds, offering a comprehensive clinical setting for ICU nursing practice and education. While Benner’s original Novice to Expert model conceptualizes the “competent” stage as typically emerging around five years of clinical practice, this study employed an adaptation of Benner’s framework as modified for the Korean clinical nursing context [24]. According to this adaptation, clinical experience levels are categorized as follows: 0–1 year (novice), 2–3 years (advanced beginner), 4–6 years (competent), and 7 years or more (proficient). Based on this model, nurses with under three years of clinical experience fall within the novice to advanced beginner stages—characterized by developing situational awareness, limited experiential depth, and a need for structured learning [24]. In Korean ICU settings, nurses in this stage often begin to practice semi-independently but continue to require ongoing support and systematic competency development. The three-year cut-off was therefore applied to focus on nurses who were in the early phase of professional growth within the ICU environment. Moreover, ICU nurses at this stage are known to experience elevated stress levels and higher turnover rates due to limited practical experience and the ongoing development of professional competency. Nurses with three or more years of ICU work experience were excluded from the study. Additionally, individuals with a history of severe adverse reactions to VR devices—such as nausea, vomiting, panic attacks, photosensitivity, or epilepsy—were excluded to ensure participant safety during the VR intervention. Participants were recruited through internal hospital announcements and direct invitations. Interested nurses contacted the research team and underwent eligibility screening. Following informed consent, participants were assigned to the experimental or control group according to ICU shift schedules and operational feasibility, minimizing disruption to patient care.

The required sample size was calculated using G*Power version 3.1.9.7, with a two-tailed test, α = 0.05, power (1 × β) = 0.80, and an effect size of f = 0.25 (medium). Based on these parameters, a minimum of 27 participants per group was required. To accommodate potential attrition, a target total sample size of 65 participants was established. To achieve this target, a total of 73 ICU nurses were screened for eligibility. Of these, seven nurses were excluded for not meeting the inclusion criteria. The remaining 66 nurses were enrolled and allocated to either the experimental group (n = 33) or the control group (n = 33), based on ICU shift schedules and operational considerations. During the study, 1 participant in the experimental group withdrew due to resignation, resulting in 32 participants completing both the intervention and follow-up assessments. All 33 participants in the control group completed the study as planned. Consequently, data from 32 participants in the experimental group and 33 participants in the control group were included in the final analysis (Figure 1).

### 2.3. Intervention: 360° VR-Based Mechanical Ventilation Nursing Education Program

#### 2.3.1. Program Development and Structure

The VR-based education program was developed following the ADDIE instructional design model (analysis, design, development, implementation, and evaluation), ensuring a systematic and learner-centered approach [25]. Analysis: An educational needs assessment was conducted through literature review and consultation with ICU nursing experts. The key learning gaps identified included managing common ventilator alarms, integrating patient assessment with ventilator management, and enhancing clinical reasoning under simulated time pressure. Design: The program aimed to immerse participants in realistic ICU scenarios using 360° VR technology, with learning objectives focused on ventilator alarm recognition and interpretation, patient assessment, and clinical reasoning. Scenario algorithms were developed for high-pressure alarms, low exhaled volume alarms, and high respiratory rate alarms. Development: The VR-based education program was developed in collaboration with a professional VR development company specializing in healthcare simulation, using the Unreal Engine 5, and implemented on Meta’s Oculus platform. The company has expertise in designing virtual environments for nursing education. Initially designed as 360° video content, the VR simulation was adapted to 3D model-based VR content to enable more dynamic interaction and scenario variation, based on pilot testing feedback and COVID-19-related clinical environment restrictions. The final VR content included key clinical situations: apnea episodes, changes in respiratory rate, variations in tidal volume, and realistic ICU background sounds to enhance immersion. The usability and fidelity of the VR environment were evaluated through iterative feedback from two nursing professors, three thoracic surgery nurses, and four ICU nurses. Their suggestions were incorporated to enhance clinical accuracy, visual realism, and interactivity.

#### 2.3.2. Implementation: The Intervention Consisted of Two Sessions

The intervention consisted of two sequential sessions delivered on the same day to accommodate ICU shift rotation constraints. For the experimental group, VR training sessions were scheduled during day shifts or off-duty periods to avoid interfering with clinical duties. Participants were permitted to voluntarily select their preferred time slots based on their personal schedules and availability. This scheduling strategy aimed to reduce fatigue related to shift work and minimize the burden of participation. Session 1 involved individualized VR-based training conducted in a dedicated simulation laboratory (average duration: approximately 17 min). The session comprised three components: (1) orientation to the VR equipment, (2) engagement with randomized ventilator alarm scenarios via an HMD, and (3) structured debriefing facilitated by the researcher, a nursing professor. The debriefing process followed the DAA (description, analysis, and application) framework to ensure consistency and depth across participants. During the description phase, participants recalled the simulated events; in the analysis phase, they critically examined their clinical decisions and nursing actions; and in the application phase, they explored how to transfer these insights to real-world ICU practice. To support critical reflection, participants reviewed their own simulation performance using a high-resolution tablet.

Session 2 consisted of one-on-one, face-to-face instruction delivered by experienced ICU nurse educators. This instruction was supplemented with printed educational materials and reference texts aligned with the VR content. The blended learning format was designed to reinforce and contextualize skills acquired during the VR session (Figure 2). The content validity of both the VR program and the educational materials was evaluated by an expert panel comprising two nursing faculty members with ICU experience, three thoracic surgery nurses specializing in ventilator care, and four ICU nurses with more than 20 years of clinical experience. This review process yielded a content validity index (CVI) of ≥0.80. Additionally, pilot testing with five ICU nurses demonstrated high levels of engagement and feasibility, leading to iterative refinements in the scenario flow and user interface design (Figure 3).

#### 2.3.3. Control Group

Participants in the control group first completed the pre-test questionnaire and then received conventional educational materials, including printed reference guides and textbooks related to mechanical ventilation nursing. To ensure consistency with standard ICU nurse education practices, no VR-based instruction was provided to this group. Following the completion of the experimental group’s intervention period, the control group’s participants completed the post-test questionnaire.

### 2.4. Measurements

All measures were administered to both groups at baseline (pre-test) and immediately following the intervention period (post-test), with the exception of the sense of presence, which was assessed only in the experimental group after the intervention. The following validated instruments were employed to assess the study outcomes:

#### 2.4.1. Knowledge of Mechanical Ventilation Nursing

Knowledge of mechanical ventilation nursing was assessed using an instrument developed by the researchers based on the American Association for Respiratory Care (AARC) clinical guidelines [26]. The instrument consists of 16 items across 5 domains: initiation of mechanical ventilation (6 items), application and management (3 items), alarm control (5 items), mechanical ventilation failure (1 item), and ventilator weaning (1 item). All items are presented in a five-option multiple-choice format. Each item is scored dichotomously (1 = correct, 0 = incorrect), resulting in a total score ranging from 0 to 16, with higher scores indicating greater knowledge. The examples of the questions are as follows: “A patient with an artificial respiratory system shows persistent hypercapnia based on arterial blood gas analysis. Which ventilator setting should be adjusted first to correct this condition?” and “If a patient on mechanical ventilation triggers a high Peak Inspiratory Pressure (PIP) alarm, what should be the nurse’s first action?” Content validity was evaluated by a panel of experts comprising two nursing faculty members and three nurses specializing in ventilator management. Items with a content validity index (CVI) of ≥0.80 were selected for inclusion in the final instrument. The KR-20 coefficient was 0.79 in this study.

#### 2.4.2. Self-Efficacy for Ventilation Nursing

Self-efficacy for ventilation nursing was measured to assess the level of confidence in performing care for patients receiving mechanical ventilation. The instrument used was originally developed by Ha and Koh [27] and later applied in a study by Lee and Han [10] for nurses caring for mechanically ventilated patients. The scale consists of 10 items rated on a 5-point Likert scale (1 = strongly disagree to 5 = strongly agree), with higher scores indicating greater self-efficacy in performing ventilation nursing tasks. The reliability of the instrument, as measured by Cronbach’s α, was 0.97 at the time of development [27], 0.78 in the previous study [10], and 0.79 in the present study.

#### 2.4.3. Clinical Reasoning

Clinical reasoning was measured using the Korean version of the Nurse Clinical Reasoning Scale (NCRS), originally developed by Liou et al. [28] and translated and adapted by Joung and Han [29]. The scale assesses the ability of ICU nurses caring for mechanically ventilated patients to collect relevant patient information and apply cognitive processes necessary for implementing the nursing process. The instrument consists of 15 items rated on a 5-point Likert scale (1 = not at all to 5 = very much so), with higher scores indicating higher levels of clinical reasoning. The reliability of the instrument, as measured by Cronbach’s α, was 0.93 in the previous study [29] and 0.93 in the present study.

#### 2.4.4. Learning Immersion

Learning immersion was measured to assess the degree of engagement experienced by ICU nurses during the educational program. The instrument used was the Korean version of the Flow Short Scale, originally developed by Engeser and Rheinberg [20] and adapted for nursing education by Yoo and Jim [30] and Lee and Han [10]. The scale consists of 10 items rated on a 5-point Likert scale (1 = not at all to 5 = very much so), with higher scores indicating a greater level of learning immersion. The reliability of the instrument, as measured by Cronbach’s α, was 0.92 at the time of development, 0.94 in the previous study [10], and 0.93 in the present study.

#### 2.4.5. Turnover Intention

Turnover intention was measured using a scale originally developed by Lawler [31] to assess nurses’ intention to leave their organization. The scale was adapted for the Korean nursing context by Park [32]. It consists of 4 items rated on a 5-point Likert scale (1 = strongly disagree to 5 = strongly agree), with higher scores indicating a stronger intention to leave. The reliability of the instrument, as measured by Cronbach’s α, was 0.88 in the previous study [32] and 0.82 in the present study.

#### 2.4.6. Sense of Presence

Sense of presence was assessed among the participants in the experimental group to evaluate the level of perceived presence during the VR intervention. The Presence Questionnaire, originally developed by Witmer and Singer [33] and subsequently revised by Lee et al. [34], was used for this purpose. The instrument consists of 19 items rated on an 11-point scale (0 = not at all to 10 = very much so), with higher total scores indicating a stronger sense of presence. The Cronbach’s alpha for this instrument was 0.94 in this study.

#### 2.4.7. Educational Satisfaction

Educational satisfaction was measured post-intervention using a single-item numeric rating scale (NRS) ranging from 0 (very dissatisfied) to 10 (very satisfied), with higher scores indicating greater satisfaction with the educational experience. In addition, qualitative feedback was collected from participants in the experimental group regarding their experiences with the VR-based education program. Participants were invited to provide open-ended comments on areas for improvement in the program as well as suggestions for future VR training topics in intensive care nursing. These qualitative responses were categorized and summarized for analysis.

### 2.5. Data Collection

Data collection was conducted over a six-month period from May 2023 to October 2023. Baseline assessments for all participants were completed prior to the implementation of the educational intervention. Following the completion of the intervention, post-test assessments were administered immediately for both the experimental and control groups to evaluate the outcomes. All VR-based simulation sessions were conducted in a dedicated simulation laboratory to ensure consistency of the learning environment. A single experienced facilitator led all debriefing sessions, providing uniform instructional quality and feedback across participants. Additionally, a trained research assistant was responsible for administering both the pre-test and post-test instruments to all participants, ensuring standardized data collection procedures.

### 2.6. Data Analysis

Data were analyzed using SPSS version 29.0 for Windows, with statistical significance set at *p* < 0.05. The normality of the dependent variables was tested using the Shapiro–Wilk normality test, and all variables met the assumption of normality. The general characteristics of participants were summarized as means and standard deviations for continuous variables and as frequencies and percentages for categorical variables. The homogeneity of the experimental and control groups in terms of general characteristics and pre-intervention measures was tested using independent-sample *t*-tests, the χ^2^-test, and Fisher’s exact test where appropriate (i.e., when expected cell frequencies were less than five). To assess the impact of the intervention within each group, pre- and post-test differences were analyzed using paired *t*-tests. To evaluate between-group differences in intervention effects, independent-sample *t*-tests were performed on difference scores (Δ score, calculated as post-test score minus pre-test score) for each dependent variable.

### 2.7. Ethical Considerations

This study was approved by the Institutional Review Board (IRB) of C University (IRB No. 2-1041055-AB-N-01-2023-18) prior to data collection and intervention implementation. Approval was obtained in accordance with the ethical principles outlined in the Declaration of Helsinki. The nursing department of the participating hospital provided organizational support for participant recruitment. Participation in the study was entirely voluntary. Written informed consent was obtained from all participants after a thorough explanation of the study’s purpose, procedures, potential risks, and benefits. Participants were explicitly informed of their right to withdraw from the study at any time without penalty or disadvantage. To ensure participant safety during the VR intervention, measures were taken to minimize physical discomfort, particularly around the eyes during HMD use. Throughout the study, no participants reported experiencing adverse effects such as nausea, vomiting, panic attacks, photosensitivity, or other VR-related side effects. The research team monitored all sessions closely to ensure participant well-being and comfort. Participant anonymity and data confidentiality were strictly maintained. All collected data were securely stored and used solely for research purposes.

## 3. Results

### 3.1. Homogeneity Test of General and Clinical Characteristics and Dependent Variables Between the Two Groups

A total of 65 ICU nurses participated in the study, with 32 in the experimental group and 33 in the control group. The groups were comparable in gender distribution, age, ICU career length, educational background, prior critical care education, and VR-related experience. This test satisfied a normal distribution and identified no statistically significant differences in any general or clinical characteristics between the groups (*p* > 0.05). Additionally, baseline scores for all dependent variables—knowledge of ventilation nursing, self-efficacy, clinical reasoning, learning immersion, and turnover intention—did not differ significantly between the groups (*p* > 0.05), confirming baseline equivalence (Table 1 and Table 2).

### 3.2. Effects of the VR-Based Education Program

Knowledge of ventilation nursing showed a greater improvement in the experimental group than in the control group. The knowledge score increased by 5.54 points in the experimental group and 1.27 points in the control group. The difference between the two groups was statistically significant (t = 12.28, *p* < 0.001). Similarly, self-efficacy for ventilation nursing also demonstrated a significantly greater increase in the experimental group (Δ = 0.94) compared to the control group (Δ = 0.15) (t = 4.72, *p* < 0.001). Clinical reasoning also improved significantly more in the experimental group (Δ = 0.76) than in the control group (Δ = 0.13) (t = 3.95, *p* < 0.001). Learning immersion showed a statistically significantly greater increase in the experimental group (Δ = 0.88) compared to the control group (Δ = 0.05) (t = 6.92, *p* < 0.001). In contrast, there was no significant difference between the two groups in turnover intention (experimental group Δ = 0.14; control group Δ = 0.20) (t = 0.45, *p* = 0.652). Educational satisfaction was significantly higher in the experimental group (M = 8.75 ± 0.84) than in the control group (M = 5.91 ± 1.35) (t = 10.19, *p* < 0.001). The sense of presence in the experimental group was also high (M = 155.13 ± 18.04), indicating effective immersion in the VR-based learning environment (Table 3).

### 3.3. Participants’ Feedback on the VR-Based Education Program

Qualitative feedback was collected from participants in the experimental group regarding their experiences with the VR-based ventilation nursing education program. Overall, participants reported positive engagement with the VR training. However, they also identified several areas for improvement. With respect to content usability, novice nurses found the ventilator mode-setting section particularly challenging, suggesting that the level of difficulty should be adjusted to better accommodate varying levels of clinical experience. Participants also expressed a desire for a broader selection of critical care scenarios, noting that the current scenario set was somewhat limited. Regarding the physical usability of the VR equipment, several participants reported discomfort when wearing glasses with an HMD, and some experienced blurring depending on the viewing angle. In addition, participants provided suggestions for future VR-based education topics in intensive care nursing. The most frequently requested topics included emergency nursing and cardiopulmonary resuscitation (CPR) (14 responses), endotracheal intubation nursing (12 responses), continuous renal replacement therapy (10 responses), extracorporeal membrane oxygenation (ECMO) management (7 responses), and advanced ventilator management (3 responses). These participant insights underscore the need for further development of VR-based educational content to address a wider range of clinical competencies and to enhance the usability of the VR learning environment (Table 4).

## 4. Discussion

This study aimed to examine the impact of a 360° VR education program on ICU nurses’ ventilator management competence, self-efficacy, clinical reasoning, learning immersion, turnover intention, and educational satisfaction within the context of Korean critical care settings. The findings demonstrated significant improvements in ventilator management knowledge, self-efficacy, clinical reasoning, learning immersion, and educational satisfaction among participants exposed to the VR intervention compared to those receiving conventional print-based education. However, no significant change in turnover intention was observed. In this discussion, we interpret these findings in light of existing research and Korea-specific ICU contexts while considering implications, limitations, and directions for future research.

### 4.1. Strengthening Theoretical Knowledge Through Immersive Simulation

The significant improvement in ventilator management knowledge observed in the experimental group is consistent with prior evidence demonstrating that VR simulation facilitates deep learning and enhances core nursing competencies in complex clinical domains. For example, Alsharari et al. [35] reported that VR-based simulation significantly improved knowledge, problem-solving skills, and professional competencies among nursing students. Similarly, several studies have demonstrated that VR-based simulation can enhance theoretical knowledge, familiarity, and confidence in managing critical care procedures such as tracheostomy care and neonatal infection control, although the extent of knowledge gains reported across different studies has been variable [36,37]. However, in a previous study applying a VR-based mechanical ventilation nursing program to 60 Korean nursing students [10], no statistically significant difference in knowledge between the intervention and control groups was observed. Notably, the immersive and high-fidelity, scenario-based VR simulation in our study—designed specifically for practicing ICU nurses and incorporating dynamic ventilator alarms and complex patient care situations—appears to have promoted greater learner engagement and deeper acquisition of theoretical knowledge. While these findings support the potential of VR-based education to improve knowledge, it is important to interpret the results within the context of the intervention’s limitations. Given the short, single-day duration of the program, the observed improvement likely reflects immediate recall and short-term knowledge acquisition rather than long-term retention or comprehensive competency development. Furthermore, the provision of user-friendly supplementary educational materials may have further enhanced learning outcomes. Given the specialized demands of ICU nursing in Korea, where opportunities for repeated and systematic learning are often limited, particularly during public health crises, immersive VR simulation offers a valuable strategy to bridge the gap between theoretical knowledge and practical competence. However, future studies with longitudinal follow-up are needed to evaluate sustained knowledge retention and performance-based outcomes.

### 4.2. Enhancing Clinical Self-Efficacy

Self-efficacy scores significantly improved in the VR group, consistent with findings from prior research. Lee and Han [10] reported that VR simulation-based ventilation training enhanced self-confidence and clinical competence among Korean nursing students. Similarly, Chan et al. [16] demonstrated positive outcomes in self-efficacy and procedural proficiency following VR-based chemotherapy simulation, as well as in ECMO nursing education [17].

It is possible that the immersive and interactive nature of the 360° VR experience temporarily boosted learners’ confidence in managing patients on mechanical ventilation. While these findings suggest the potential of VR simulation to enhance clinical self-efficacy, the short duration of the intervention limits the extent to which these improvements can be considered sustainable. In the Korean ICU context, where nurses frequently encounter high-pressure environments, intricate equipment, and alarm fatigue, fostering self-efficacy in ventilator management is particularly critical. Enhanced self-efficacy not only promotes critical thinking and proactive clinical responses but also contributes to a culture of safety and may ultimately improve patient care outcomes.

### 4.3. Strengthening Clinical Reasoning

Participants who underwent VR training exhibited greater improvements in clinical reasoning than the control group. While complex decision-making under simulated ventilator alarms—requiring real-time interpretation and rapid response—may have helped enhance reasoning skills more effectively than traditional lectures or static study materials [19], this finding should be interpreted with caution. Given the short duration of the intervention, the observed improvements may reflect short-term gains in reasoning performance rather than sustained development of clinical decision-making capacity. This aligns with Padilha et al. [18], who demonstrated enhanced clinical reasoning in nursing students following clinical virtual simulation, and Salameh et al. [38], who reported improved critical thinking among mechanical ventilation students post high-fidelity simulation. Our findings suggest that 360° VR scenarios may serve as a practical and scalable educational tool to strengthen clinical reasoning in the high-demand context of Korean ICUs, where staffing shortages and elevated patient acuity require advanced cognitive and decision-making skills. Nonetheless, future longitudinal studies are needed to assess whether such improvements persist over time and translate into real-world clinical performance.

### 4.4. Learning Engagement and Immersion as Catalysts

The enhanced learning immersion observed among experimental group participants reinforces the unique value of 360° VR. Immersion and presence are widely recognized as critical facilitators of effective learning, fostering deeper cognitive and emotional engagement [13]. Multiple studies involving Korean nursing students have demonstrated that VR-integrated learning environments significantly improve engagement and motivation compared to traditional methods [39]. In high-stress clinical contexts such as the ICU, where sustaining attention can be particularly challenging, immersive VR environments help mitigate cognitive fatigue by offering a safe yet realistic space for practice. Participants in this study reported high levels of perceived presence, indicating that the VR scenarios effectively created a sense of “being there”—a factor closely associated with improved knowledge retention and skill acquisition. In the context of Korean ICU training, where opportunities for hands-on experience are often constrained by equipment limitations and staffing shortages, integrating immersive VR represents a highly promising approach to enhancing the quality and accessibility of clinical education. These findings support the broader integration of VR-based training in ICU nursing education to complement conventional methods and address existing educational gaps.

### 4.5. Educational Satisfaction and Perceived Utility

The significantly higher educational satisfaction among VR-trained nurses underscores the subjective value of such interactive and contextually relevant simulation. Findings from several systematic reviews have similarly reported that participants generally prefer immersive VR-based educational interventions [3,13,40]. In our study, participants described the VR experience as engaging, realistic, and directly applicable, thereby reinforcing its perceived relevance to real-world clinical practice. Higher educational satisfaction may, in turn, enhance motivation and increase the likelihood that learners will apply acquired skills in actual clinical settings. In Korean ICUs, where nurse turnover is high and burnout is prevalent [23], fostering satisfaction through effective training interventions may contribute to sustained professional engagement, even if no immediate reduction in turnover intention was observed in this study.

### 4.6. Exploring Workforce Retention: Turnover Intention

No statistically significant change in turnover intention was observed. Turnover in ICU nursing is a global concern, driven by workload, emotional stress, moral distress, leadership, work–life balance, and compensation [22]. While improvements in educational quality and self-efficacy may contribute to job satisfaction, these factors alone may not be sufficient to influence turnover intention in the short term. Previous studies emphasized the multifactorial nature of nurse retention, suggesting that structural changes, supportive leadership, and team dynamics are crucial [22,23]. Moreover, baseline scores for turnover intention were relatively low in both groups, which may have resulted in a floor effect, limiting the capacity to detect meaningful changes following the intervention. This limitation should be considered when interpreting the non-significant findings. Future studies incorporating participants with more diverse baseline turnover intention levels or adopting longer follow-up periods may be necessary to more accurately assess the impact of VR-based education on workforce retention outcomes. Given these complexities, while VR training may enhance critical-care competency and reduce some stressors, affecting turnover intention likely requires broader organizational strategies. Longitudinal studies that measure actual retention rates following VR-based interventions may provide further insight into the role of education in promoting workforce stability.

### 4.7. Korea-Specific ICU Context

Internationally, VR-based education is gaining traction; however, implementation in Korean ICUs remains nascent. Cultural norms emphasizing hierarchical decision-making, rigid work hours, and high expectations of self-reliance present challenges [23]. VR training programs designed for Korea and embedded within institutional continuous professional development systems may accelerate adoption. The high presence ratings and enthusiastic feedback—even in a conservative clinical culture—indicate openness to technological solutions. Moreover, participants’ recommendations for hardware comfort and scenario expansion (e.g., ECMO, CPR, and CRRT) reflect intrinsic motivation toward competency development. Hospitals could mainstream VR simulation kits into regular training cycles and align content with national ICU nursing competencies.

### 4.8. Feedback and Curriculum Optimization

Qualitative feedback highlighted the necessity to adjust VR-induced difficulty, especially for novice nurses less skilled in ventilator mode settings. Incorporating adaptive learning layers—whereby difficulty increases as competence develops—could enhance the program’s responsiveness to learners’ needs. Participants also expressed strong interest in expanding scenario types to include emergency procedures and complex equipment. Including regularly updated modules aligned with changing ICU protocols (e.g., prone ventilation, advanced ventilator modes) could significantly enhance clinical relevance and upgrade professionals’ skill sets in line with evolving ICU roles in Korea. The hardware concerns raised—such as optometric discomfort and visual blurring—underscore practical barriers to adoption. Ongoing advancements in VR hardware—such as improved ergonomics, reduced visual strain, and enhanced compatibility with prescription glasses—could address these limitations and enhance usability, thereby facilitating wider integration of VR-based education in ICU settings.

### 4.9. Implications for Practice

This study demonstrates that the 360° VR-based mechanical ventilation nursing education program is an effective educational strategy for enhancing the competencies of ICU nurses in Korea. Based on these findings, several specific recommendations for practical application are proposed. First, considering that education for ICU nurses in many tertiary and university hospitals currently relies heavily on theoretical lectures and skills-based training, it would be highly desirable to formally integrate VR-based simulation modules into the structured education pathways for both novice ICU nurses and experienced nurses requiring competency updates [41,42]. Second, although this study confirmed the short-term efficacy of the program, it is necessary to provide regular, periodically updated VR scenario-based training to maintain and further develop competencies related to ventilator alarm management and advanced ventilator operation. Regular refreshers (e.g., semi-annual or annual VR simulation sessions) would support sustained learning and clinical performance. Third, while the VR content developed in this study reflects commonly encountered alarm management situations, customization of VR scenarios to reflect hospital-specific protocols and equipment is highly recommended. Since variations in clinical practice and ventilator models exist across institutions, tailored VR content could further enhance training relevance and applicability. Fourth, the educational effects demonstrated in this study suggest strong potential for expanding VR-based training to other high-risk clinical environments, such as emergency departments, cardiovascular centers, and operating rooms, where mechanical ventilation management is also critical [42]. Fifth, although no significant short-term changes in turnover intention were observed in this study, improvements in self-efficacy, clinical reasoning, and educational satisfaction may indirectly contribute to enhanced job satisfaction and retention. Therefore, incorporating VR-based education into workforce support and burnout prevention strategies should be considered by hospital administrators and nursing leadership. Additionally, integrating VR-based training with interprofessional ICU team training could foster collaborative ventilator management practices and improve interdisciplinary communication [36]. Finally, it is essential to establish a systematic evaluation framework to assess the long-term clinical impact of VR-based education. Future initiatives should include tracking the application of VR-acquired skills in clinical practice, monitoring patient safety indicators, and analyzing the contribution of VR training to overall ICU performance, patient safety, and staff well-being. Furthermore, embedding 360° VR simulation as a component of mandatory ICU continuing education could contribute to standardizing ventilator management competencies across healthcare institutions in Korea. However, despite its immersive and flexible nature, VR-based education also presents certain limitations. These include the lack of tactile feedback, the potential for user discomfort (e.g., cybersickness), and dependence on learners’ digital literacy. Moreover, its educational effectiveness may vary depending on the realism of the simulated scenarios, the availability and quality of an HMD, and faculty members’ familiarity with immersive instructional technologies.

### 4.10. Limitations and Future Directions

Despite its strengths, this study has limitations. First, the quasi-experimental design and non-random group allocation may introduce selection bias, though baseline homogeneity supports group equivalence. Second, the sample was restricted to a single tertiary hospital, which may limit generalizability. Multi-center randomized controlled trials are warranted to validate these findings across diverse healthcare contexts in Korea. Third, all outcome variables—including knowledge, clinical reasoning, self-efficacy, and turnover intention—were measured immediately after the intervention. Without long-term follow-up, it is unclear whether the observed improvements represent temporary learning gains or sustainable behavioral change. The durability of these gains—particularly in real-world clinical performance and patient outcomes—requires longitudinal evaluation. Fourth, the study did not include objective performance-based indicators such as direct observation, response time metrics, or error tracking in ventilator alarm management. Reliance on self-reported outcomes may overestimate perceived competence and does not fully capture actual clinical performance. Future studies should incorporate validated, performance-based outcome measures to assess the real-world impact of VR-based education. Fifth, turnover intention was assessed only once, immediately post-intervention. Longer-term follow-up could reveal different patterns over time, particularly as learners integrate new skills into clinical routines and organizational environments evolve. Additionally, the use of a single-item scale to measure educational satisfaction may limit the depth and nuance of assessment. Future studies are recommended to employ multi-item, validated instruments that can capture various dimensions of learner satisfaction. Moreover, qualitative data from participants offer valuable insight into program optimization. Future research should apply mixed-methods approaches to capture long-term clinical application, confidence retention, interprofessional collaboration, and integration of VR-acquired skills into routine ICU practice. To enhance external validity and practical relevance, future studies should also investigate how VR training influences measurable clinical outcomes such as alarm management accuracy, task efficiency, and patient safety indicators.

## 5. Conclusions

The 360° VR-based mechanical ventilation nursing education program demonstrated significant short-term improvements in ICU nurses’ knowledge, clinical reasoning, self-efficacy, learning immersion, and educational satisfaction compared to conventional print-based education. These findings highlight the potential of immersive VR simulation as a supplementary educational strategy in high-acuity ICU environments, where opportunities for hands-on training are often limited. This is particularly relevant in the Korean healthcare education context, where the implementation of high-fidelity simulation programs is often constrained by space, budget, and personnel. In contrast, VR-based education requires relatively less infrastructure and operational support, making it a more scalable and accessible option for clinical training. The program also demonstrated strong feasibility and acceptability among practicing ICU nurses, as reflected in high satisfaction and presence scores. Although no significant change was observed in turnover intention, the enhancements in competence and learning experience may indirectly support nurse retention when integrated within broader workforce development strategies.

Nonetheless, the current findings should be interpreted within the context of certain limitations, including the short duration of the intervention and the absence of objective performance-based measures or long-term follow-up. Future studies are warranted to examine the sustained impact of VR-based education on clinical performance, patient outcomes, and nurse retention through longitudinal, multi-center randomized controlled trials. Additionally, expanding VR content to encompass a wider range of ICU competencies—such as ECMO, CRRT, and CPR—and leveraging ongoing advancements in VR hardware will further enhance its usability and applicability. In the Korean ICU context, characterized by high patient acuity and limited simulation resources, integrating 360° VR into national ICU nurse education frameworks holds strong potential to advance critical care nursing excellence. As healthcare demands and critical care complexity continue to evolve, VR-based learning may serve as a forward-looking, scalable, and context-sensitive approach to supporting innovation in ICU nursing education and practice.

## Figures and Tables

**Figure 1 healthcare-13-01639-f001:**
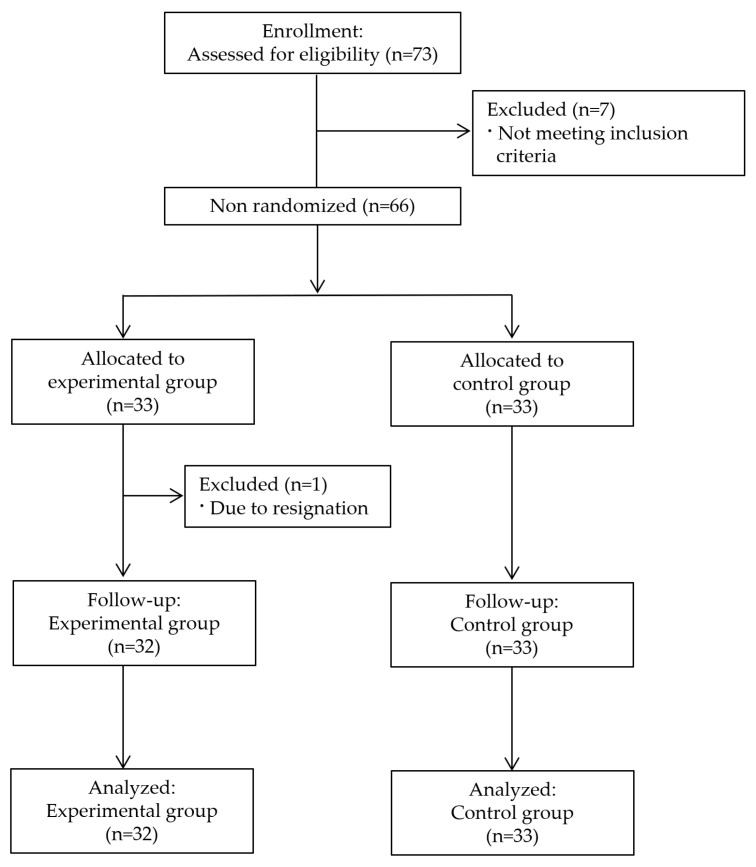
Flowchart of participant selection throughout the study.

**Figure 2 healthcare-13-01639-f002:**
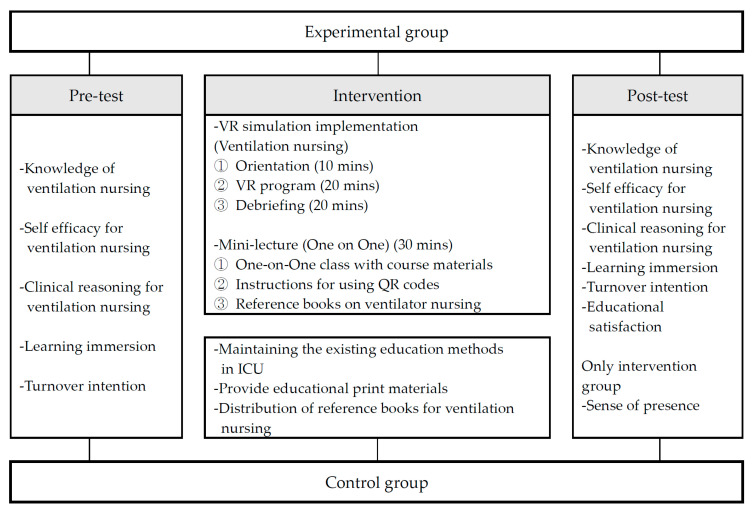
Research design.

**Figure 3 healthcare-13-01639-f003:**
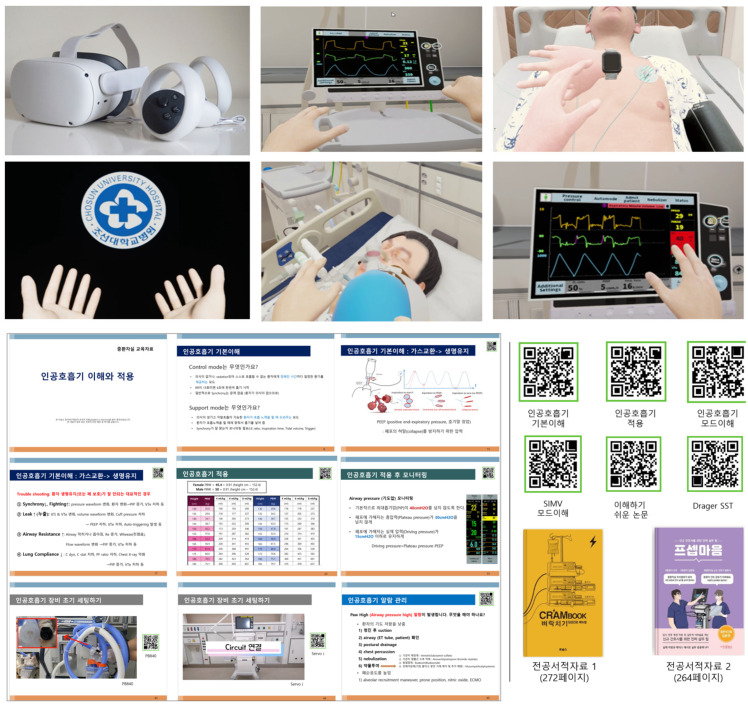
Representative scenes and educational materials from the VR intervention.

**Table 1 healthcare-13-01639-t001:** Homogeneity of general and clinical characteristics between the experimental and control groups (N = 65).

Variables	Experimental Group (n = 32)	Control Group (n = 33)	t or X^2^	*p*
	n (%), Mean ± SD			
Gender				
male	4 (12.5)	7 (21.2)	0.877	0.349
female	28 (87.5)	26 (78.8)		
Age (years)	25.41 ± 2.79	25.21 ± 1.69	0.340	0.735
Career (months)	13.93 ± 8.20	18.66 ± 11.16	1.950	0.056
Education				
associated degree	6 (18.8)	8 (24.2)	0.290	0.590
bachelor	26 (81.2)	25 (75.8)		
graduate	0 (0.0)	0 (0.0)		
Completion of critical care coursework				
in undergraduate school				
yes	22 (68.8)	19 (57.6)	0.871	0.351
no	10 (31.2)	14 (42.4)		
Completion of critical care practicum				
in undergraduate school				
yes	18 (56.3)	22 (66.7)	0.745	0.388
no	14 (43.8)	11 (33.3)		
Attended additional educational				
course related to ventilation nursing				
yes	7 (21.9)	9 (27.3)	0.255	0.614
no	25 (78.1)	24 (72.7)		
Experience VR devices for education				
yes	5 (15.6)	6 (18.2)	0.076	0.783
no	27 (84.4)	27 (81.8)		
Experience VR devices for entertainment				
yes	12 (37.5)	11 (33.3)	0.123	0.725
no	20 (62.5)	22 (66.7)		

Note: SD = standard deviation; VR = virtual reality.

**Table 2 healthcare-13-01639-t002:** Homogeneity of baseline dependent variables between the experimental and control groups (N = 65).

Variables	Experimental Group (n = 32)	Control Group (n = 33)	t	*p*
	Mean ± SD			
Knowledge of ventilation nursing	9.09 ± 1.80	8.88 ± 2.45	0.402	0.689
Self-efficacy for ventilation nursing	2.89 ± 0.61	3.13 ± 0.45	1.781	0.080
Clinical reasoning for ventilation nursing	2.93 ± 0.47	3.12 ± 0.51	1.561	0.124
Learning immersion	3.14 ± 0.72	3.23 ± 0.42	0.608	0.546
Turnover intention	2.92 ± 0.93	2.95 ± 0.73	0.158	0.875

Note: SD = standard deviation.

**Table 3 healthcare-13-01639-t003:** Effects of the VR education program.

Variables	Time	Experimental Group (n = 32)	Control Group (n = 33)	T (*p* ^1^)
		Mean ± SD		
Knowledge of ventilation	Pre-test	9.09 ± 1.80	8.88 ± 2.45	12.284 (<0.001)
nursing	Post-test	14.63 ± 1.24	10.15 ± 1.66	
	T (*p* ^2^)	15.051(<0.001)	2.981(0.005)	
Self-efficacy	Pre-test	2.89 ± 0.61	3.13 ± 0.45	4.719
for ventilation nursing	Post-test	3.83 ± 0.41	3.28 ± 0.51	(<0.001)
	T (*p* ^2^)	8.301(<0.001)	2.215 (0.034)	
Clinical reasoning	Pre-test	2.93 ± 0.47	3.12 ± 0.51	3.946
for ventilation nursing	Post-test	3.69 ± 0.39	3.25 ± 0.49	(<0.001)
	T (*p* ^2^)	9.365(<0.001)	1.579 (0.124)	
Learning immersion	Pre-test	3.14 ± 0.72	3.23 ± 0.42	6.915
	Post-test	4.02 ± 0.47	3.28 ± 0.39	(<0.001)
	T (*p* ^2^)	6.148(<0.001)	0.577 (0.568)	
Turnover intention	Pre-test	2.92 ± 0.93	2.95 ± 0.73	0.453
	Post-test	3.06 ± 0.94	3.15 ± 0.61	(0.652)
	T (*p* ^2^)	1.206(0.237)	1.831 (0.076)	
Education satisfaction	Post test	8.75 ± 0.84	5.91 ± 1.35	10.186
				(<0.001)
Sense of presence	Post test	155.13 ± 18.04	N/A	N/A

Note: *p*
^1^ = *p*-value by the independent-sample *t*-test, *p*
^2^ = *p*-value by paired *t*-test, N/A = not applicable.

**Table 4 healthcare-13-01639-t004:** Participants’ feedback on the VR-based education program.

**Improvement Needs After Participating in a VR Training Program**
- The section for setting the mode on the ventilator proved challenging for novice nurses. Therefore, the level of difficulty requires adjustment. - It would be advantageous to have a wider range of critical care scenarios available instead of the current limited selection. - Wearing glasses with the VR HMD caused discomfort. - There was some blurring when using a VR HMD, depending on the angle.
**Topic Needs for VR Education Program Development in Intensive Care Nursing**
- Emergency nursing and cardiopulmonary resuscitation (CPR) (14 responses) - Endotracheal intubation nursing (12 responses) - Continuous renal replacement therapy (10 responses) - Extracorporeal membrane oxygenation (ECMO) management (7 responses) - Advanced ventilator management (3 responses)

## Data Availability

The raw data supporting the conclusions of this article will be made available by the authors upon request.

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
