# Peer review of "The Effectiveness of 360-Degree Virtual Reality-Based Mechanical Ventilation Nursing Education for ICU Nurses"

_healthcare, 2025, doi:10.3390/healthcare13141639_

Round 1
Reviewer 1 Report
Comments and Suggestions for Authors
Dear authors, thank you for allowing me to review this interesting manuscript titled "Effectiveness of 360-Degree Virtual Reality-Based Mechanical Ventilation Nursing Education for ICU Nurses: A Quasi-Experimental Study". The study addresses a relevant and timely topic in the field of critical care nursing education. The implementation of immersive 360° virtual reality (VR) to support ventilator training in ICU nurses—rather than students—is novel and commendable. The study is methodologically sound, and the manuscript is clearly written. However, several issues require clarification or revision to strengthen the validity and generalizability of the findings.
I list my comments here below:
ABSTRACT
- Please define the Δ symbol at first use for clarity.
- Indicate explicitly that the post-test assessments were performed immediately after the intervention.
- Consider rephrasing the impact on turnover intention more cautiously, as no significant change was observed.
INTRODUCTION
- The theoretical framework, which references Benner’s model, is problematic. Benner generally defines the "competent" level at around 5 years of experience, not 3. The justification for using a 3-year cut-off should be clarified or reframed.
- The novelty of this study compared to prior VR studies (e.g., Lee & Han, 2022) should be emphasised more clearly.
- The introduction could be more concise by reducing some redundancies regarding the challenges of ICU nursing and mechanical ventilation.
METHODS
- Nature of the VR platform: It is unclear whether the virtual environment was developed in-house or adapted from commercial software. Please clarify the following: What software or engine was used? Was it developed by the research team or through a third-party company? Was there any validation or testing of its usability and fidelity?
- The duration and structure of the intervention are not clearly defined. Specifically: What was the time interval between Session 1 and Session 2? Was the entire intervention completed in a single day?
- Debriefing process: Provide more information on the debriefing method used (e.g., PEARLS, GAS, or other frameworks), who conducted it, and whether it followed a standardized structure.
- Please consider including sample items or supplementary material for the custom-developed knowledge test, as this enhances transparency and reproducibility.
- The use of a single-item scale to measure educational satisfaction is a methodological limitation and should be acknowledged.
RESULTS
- While the statistical analysis is appropriate, the emphasis on outcome improvement (e.g., clinical reasoning) should be balanced with the very brief duration of the intervention.
- Please provide more critical commentary on the potential floor effect for turnover intention, as baseline scores were already low.
- Table 3 is comprehensive but dense; consider simplifying or breaking it into sub-tables for readability.
DISCUSSION
- The manuscript overstates the effectiveness of a single, short educational event. It is unlikely that lasting improvements in complex skills, such as clinical reasoning or self-efficacy, can result from a single brief VR session (approximately 17 minutes).
- Please moderate the tone and avoid overgeneralizing results.
- More critical discussion is needed regarding the lack of long-term follow-up, the absence of objective performance-based outcomes, and the fact that improvements were measured only immediately after the intervention.
LIMITATIONS
The authors should explicitly acknowledge that:
- The intervention was short-term and not repeated.
- Outcomes were measured using self-report instruments.
- The software’s development process, cost, and reproducibility are not described, which limits its scalability.
- The study was conducted in a single centre, limiting generalizability.
CONCLUSIONS
The conclusion is generally well aligned with the findings, but could be more cautious. Please avoid statements implying general implementation readiness. Reinforce the need for longitudinal studies, multi-centre RCTs, and objective measures (e.g., clinical error rates or decision-making accuracy) to confirm the program’s real-world effectiveness.
Author Response
Please see the attachment.
Thank you for your valuable feedback. In response to the constructive feedback from the reviewers, we have thoroughly revised the manuscript to address all comments and suggestions.

Reviewer 2 Report
Comments and Suggestions for Authors Congratulations to the authors for their excellent work. The topic is innovative and the article is well-articulated and well-founded. I would like to make a few considerations: The title is to long, I suggest changing: Effectiveness of 360-Degree Virtual Reality-Based Mechanical Ventilation Nursing Education for ICUABSTRACT: Include data collection period, statistics and data analysis method.
INTRODUCTION Between line 60, it is necessary to delve deeper into what 360 virtual reality is, defining stages and concepts.
I am curious, because in several countries 360 virtual reality is more financially expensive than compared to realistic simulation mannequins. Is this the reality in Korea? Could you discuss this in the study justification?
METHOD: Justify why this hospital was selected. Regarding the participants, was only 1 ICU selected? What was the bed capacity of this ICU? Include. RESULTS: Well-founded and well-articulated. DISCUSSION: I suggest including the limitations of the use of virtual reality technology and its use in nursing education. CONCLUSION Ok There are 35% of old citations, I suggest updating citations for articles published less than 3 years ago.
Author Response

(The authors gave the same response as above.)

Reviewer 3 Report
Comments and Suggestions for Authors
Abstract is well written and clear.
Introduction: Line 34, suggest changing 'demanded of ICU nurses' to 'required of ICU nurses' or One of the essential competencies' or Among the core competencies expected of ICU nurses
Line 48-49 - suggest adding 'real life' or 'realistic' somewhere in this sentence. 'unable to replicate the realistic complexities..' as the point is that while these learning approaches are helpful in teaching the students the basics and the content, they can't replicate the myriad scenarios they may deal with in practice.
Line 51-54. suggest updating this to reflect that the pandemic may have changed or added complexities to ventilated patients based on how these patients responded to the virus. Is COVID still impacting nurses' access to learning environments? This sentence may need to be reworked to reflect the current state of health care learning environments or if this issue is that nurses who should have been learning these skills in a clinical environment missed this opportunity and now need to catch up, or if the point is that we should be prepared with a backup plan for teaching and learning if a similar situation occurs. The way it reads, it sounds like COVID is still keeping nurses from learning in a clinical environment, which may need to be updated to reflect that when this happened, we weren't prepared with an alternative plan to train ICU nurses away from bedside learning.
What is the cost difference between VR based education and high-fidelity simulation? Is the cost difference the reason that high-fidelity simulation often isn't available or widely used?
Methods:
Line 111 - what is the typical requirement of experience as and RN to work in the ICU? Some hospitals require at least 1 year of experience and/or experience in a certain setting like a med surg unit. Was the cut off three years total experience as a nurse, or just 3 years in the ICU?
Line 134 explain what the control does compared to the experimental group. Do they just take the questionnaires? This information is listed later, but suggest adding here that the control group receives conventional education materials. What shift was chosen to offer the VR to and how was this determined? Could this impact the results?
Were the control group participants offered the opportunity to experience the VR education after the study conclusion? This may be beneficial as the study showed some differences between the traditional training and the VR training. As mentioned, a longitudinal study may show additional insights
Author Response

(The authors gave the same response as above.)

Round 2
Reviewer 1 Report
Comments and Suggestions for Authors
Dear Authors, thank you for allowing me to review this second version of the manuscript. I appreciate that you have provided a point-by-point response to all my comments, and I am satisfied with the answers. I congratulate you on having improved the overall quality of your manuscript.